# Past, Present and Future Marine Microwave Satellite Missions in China

Mingsen Lin [1,2] and Yongjun Jia [1,2,3,*]

1    National Satellite Ocean Application Service, Beijing 100081, China; mslin@mail.nsoas.org.cn
2    Key Laboratory of Space Ocean Remote Sensing and Application (MNR), Beijing 100081, China
3    Southern Marine Science and Engineering Guangdong Laboratory (Guangzhou), Guangzhou 511458, China
*    Correspondence: jiayongjun@mail.nsoas.org.cn

**Abstract:** Over the past 60 years, China has made fruitful achievements in the field of ocean microwave remote sensing satellite technology. A long-term plan has now been formulated for the development of Chinese ocean satellites, as well as the construction of a constellation of ocean dynamic environmental and ocean surveillance satellites. These will gradually form China's ocean monitoring network from space, thereby playing important roles in future ocean resource and environmental monitoring, marine disaster prevention and reduction, and global climate change. In this review manuscript, the developmental history of ocean microwave satellites and the development status of oceanic microwave remote sensing satellites in China are reviewed. In addition, China's achievements in the field of oceanic microwave remote sensing satellite technology are summarized, and the future development of China's ocean microwave remote sensing satellite program is analysed.

**Keywords:** satellites; remote sensing; radar altimetry; scatterometer

## 1. Introduction

Oceanic satellite remote sensing refers to the use of remote sensors for long-distance non-contact observations of marine areas. It has the unique advantage of large-scale, all-day, all-weather real-time imaging observations, and is an important means for global marine environmental monitoring. Through ocean satellite remote sensing, the processes from data to information and from information to knowledge can be obtained. The international application of ocean satellite remote sensing began in 1960 [1]. The United States successfully launched the world's first meteorological satellite "TIROS-1" [2] and obtained the sea surface temperature field from an altitude of approximately 700 km. This achievement marked the beginning of the development of oceanographic research using satellite data. In 1978, the United States launched two satellites, "Nimbus-7" [3,4] and "NOAA-1" [5], for the purpose of obtaining satellite observations of global marine ecological environments and dynamic environments using various frequency sensors, respectively. This was essentially the prelude to marine satellite remote sensing.

However, the use of marine remote sensing technology in China started late. During the 1960s [6], airborne infrared thermometers and airborne laser wave height meters were developed and tested at sea. Then, in the 1970s, meteorological satellite data from the United States and Japan were received for marine meteorological analyses and sea ice observations [7,8]. During the 1980s, China invested a great deal of human and material resources into the development of marine remote sensing technology and achieved major technological breakthroughs. In October of 1983, the State Oceanic Administration, Ministry of Aerospace Industry, and the Chinese Academy of Sciences jointly completed the "Report on the Development of Marine Satellite Project" and the "Special Report on the Comprehensive Demonstration of Marine Satellite Technology Economy," which were submitted to the State Council. Since then, China's marine satellite work has entered the stage of basic research and technical preparation. On 7 September 1988, China's first

meteorological satellite with two ocean-colour channels (FY-1) was successfully launched. In the 1990s, the development of marine remote sensing technology reached a new level. In May of 1996, the General Satellite Department of the State Oceanic Administration was established. And in December of the same year, a comprehensive technical and economic demonstration of an oceanic colour satellite passed the review, and was listed in the national "Ninth Five-Year" satellite launch plan. Moreover in 1997, China successfully achieved the independent reception of SeaWiFS oceanic colour satellite data of the United States, which provided high-quality, near real-time satellite data for the research and application of marine remote sensing technology in China. The white paper China Aerospace (issued in November of 2000) made it clear that the oceanic satellite series was an important part of China's long-term and stable satellite Earth observation system. In 2001, China established the National Satellite Ocean Application Service. This was another step in the construction of China's remote sensing satellite application system, followed by the establishment of the Meteorological Satellite Application Center and the Resource Satellite Application Center, and marked the formation of a framework for the integrated development of China's remote sensing satellite system. In 2002, China launched its first oceanic colour satellite (HY-1A), which symbolized that the development of marine remote sensing technology in China would enter a new stage and accomplish the development of marine satellites from scratch. Then, on 11 April 2007, China's second marine satellite (HY-1B) was successfully launched, which signified the transition of oceanic colour satellites from experimental application types to operational service types. On 16 August 2011, China's first oceanic dynamic environment satellite (HY-2A) was successfully launched. The performance of the satellite reached the international advanced level [9], filling the gap in China's real-time acquisition of oceanic dynamic environment information. Subsequently, on 10 August 2016, the "GF-3" satellite was successfully launched at Taiyuan Satellite Launch Center, which was mainly used for marine applications [10,11]. As a result, the gap in China's independent high-resolution multi-polarized SAR marine remote sensing data acquisition was successfully filled.

In this introduction, the development history of oceanic microwave satellite remote sensing technology during the past 70 years is reviewed. Then, the new achievements in the field of marine satellite remote sensing technology in China are introduced. Finally, the future development of China's marine microwave remote sensing satellite technology is analysed.

## 2. Development of Marine Satellites in China

According to the overall planning policies, China will develop and build three series of marine satellite constellations: (1). The oceanic colour satellite constellation; (2). The marine dynamic environmental satellite constellation; and (3). The marine surveillance satellite constellation. The oceanic colour satellite constellation also referred to as HY-1 series satellites, mainly focuses on the observations of suspended sediment, seawater transparency, and chlorophyll concentrations in ocean areas. The development of this series was considered to be beyond the scope of this review [12,13]. The marine dynamic environmental satellite constellation, also known as the HY-2 series satellites, mainly aims at the observations of sea surface wind fields, waves, sea surface heights, sea surface temperatures, and other ocean dynamic environmental elements [14]. The marine surveillance satellite constellation, which is also referred to as the HY-3 series satellites, mainly aims at the observations of sea surface targets, waves, sea surface wind fields, and internal waves. The progress and development plans of the HY-2 series and HY-3 series satellites are the focus of discussion in this review.

### 2.1. Progress History of China's Marine Dynamic Environmental Satellites

The HY-2A satellite was China's first polar orbit marine microwave (marine dynamic environment) satellite [14,15]. It was successfully launched on 16 August 2011 and is still operating in orbit. The HY-2A satellite has a sun-synchronous orbit with an inclination

of 99.34 degrees. The local time of the descending node is 6:00 am. In the early life of the satellite, a recursive frozen orbit with a repetition period of 14 days for multi-disciplinary ocean observations had been adopted, with an altitude of 971 km, a cycle of 104.46 min, and a daily operation of 13 + 11/14 cycles a day. However, during the later period of its life, a recursive orbit with a repetition period of 168 days for geodetic applications was adopted, with an altitude of 973 km, a cycle of 104.5 min, and a daily operation of 13 + 131/168 cycles a day.

The HY-2A satellite integrates active and passive microwave remote sensors for high-precision orbital measurements, orbit determinations, and all-weather/all-day global detection processes. The main payloads of the satellite include a radar altimeter, microwave scatterometer, and a scanning radiometer, as well as a calibration microwave radiometer (CMR) to correct the wet tropospheric path delays of the radar altimeter. The main mission goals of the HY-2A satellite are to monitor and investigate marine environments; obtain a variety of marine dynamic environmental parameters (such as sea surface wind fields, significant wave heights, ocean currents, and sea surface temperatures); directly provide measured data for forecast global climate changes, air-sea interactions, and catastrophic sea states etc. [12,14–16]. As a result, support services are offered for marine disaster prevention and mitigation, protection of marine rights and interests, development of marine resources, protection of marine environments, and marine scientific research [14–16].

The HY-2A satellite is one of the most complex earth remote sensing satellites in China. The active and passive microwave remote sensors observe the Earth at the same time. The electronic compatibility is complicated, and there are 16 antennas. Moreover, the satellite has the highest precision orbital determination ability among China's remote sensing satellites. By means of three high-precision orbital determination methods (for example, GPS, Doppler orbit determination system (DORIS), and laser ranging, the orbital determination precision of the HY-2A satellite can reach 2 cm. Although the HY-2A is essentially a scientific research satellite, its data products have been able to meet the needs of such marine applications as accurately identifying typhoons and tsunamis through monitoring, and significantly improving the timeliness of marine disaster forecasting. Therefore, the HY-2A satellite has been widely used in many fields [14,15,17–19].

The HY-2B satellite is China's second polar orbit marine microwave (marine dynamic environment) satellite, which was successfully launched by the Long March-4B rocket at the Taiyuan Satellite Launch Center on 25 October 2018 [20–22]. The HY-2B satellite is equipped with six payloads, including a radar altimeter, a microwave scatterometer, a scanning microwave radiometer, a calibration radiometer, a data collection system (DCS), and an automatic identification system (AIS). The radar altimeter is mainly used to measure sea surface heights, significant wave heights, and gravitational fields. The microwave scatterometer is used to observe the global sea surface wind fields. The scanning microwave radiometer is utilized to observe sea surface temperatures, water vapor content levels, liquid water, and rainfall intensities. The satellite's calibration radiometer is used to provide atmospheric wet tropospheric path delay corrections for the radar altimeter. The DCS receives buoy survey data in the coastal areas of China and other sea areas, and the AIS provides services for marine disaster prevention and mitigation, as well as marine fishery production activities. The data obtained by the HY-2B satellite have been found to be stable and continuous, and the data accuracy has been significantly improved when compared with the HY-2A satellite. The accuracy of the data regarding the sea surface heights, significant wave heights, and sea surface wind fields (wind speeds and directions) is similar to other orbital satellites, such as Jason-3 [23,24], and the accuracy of the sea surface temperature data has been confirmed to be close to that of international satellite systems [6].

The radar altimeter is an active microwave remote sensor with a main objective to measure sea surface height, which lays the foundation for long-term ocean monitoring from space to an extent that will ultimately lead to improve understanding of the ocean's role in global climate change [25]. The main instrument parameters are listed in Table 1.

**Table 1.** Main parameters of the HY-2A/B radar altimeter.

| Parameter | Value |
|---|---|
| Frequency | 13.58 & 5.25 GHz |
| Pulse-limited footprint | <2 km |
| Frequency bandwidth | 320 MHz |
| PRF | 2 KHz |

The microwave scatterometer is dedicated to determine the wind vector field (including wind speed and direction) above the ocean surface. Its swath is about 1750 km and can cover above 90% of global open sea area within one day. The scatterometer adopts two pencil beams to measure the backscatter energy. To meet the requirements of wind vector retrieval with high precision and a wide swath, the following scatterometer specifications were proposed, as listed in Table 2 [26,27].

**Table 2.** Main parameters of the HY-2 scatterometer.

| Parameter | Value |
|---|---|
| Frequency | 13.256 GHz |
| Transmit power | 120 W |
| Pulse width | 1.5 ms |
| Swath | 1350/1750 km |
| Polarization | HH/VV |
| Look angle | 34.8°/40.8° |
| Incidence angle | 41°/48° |
| Scanning mode | conically scanning |
| $\sigma_0$ measurement accuracy | 0.5 dB |
| $\sigma_0$ measurement range | −40~+20 dB |
| Wind cell resolution | 25 km |
| Wind speed accuracy | <2 m/s |
| Wind direction accuracy | <20° |
| Mission lifetime | 3 years |

The scanning microwave radiometer operated on HY-2A/B is a multi-channel radiometer (RM). Intended to obtain ocean parameters such as sea surface temperatures, sea surface winds, total water vapor and cloud liquid water content under all-weather conditions, the HY-2A/B RM is designed as a nine-channel instrument capable of receiving both horizontally and vertically polarized radiation, except on the 23.8 GHz channel, which only works with vertical polarization. The HY-2A/B satellite's RM instrument specification is listed in Table 3 [28,29].

**Table 3.** Main parameters of the HY-2 RM.

| Parameter | Value | | | | |
|---|---|---|---|---|---|
| Frequency (GHz) | 6.6 | 10.7 | 18.7 | 23.8 | 37.0 |
| Polarization | V H | V H | V H | V | V H |
| Scan width | | | 1600 km | | |
| Footprint size(km) | 100 | 70 | 40 | 35 | 25 |
| Sensitivity (K) | <0.5 | <0.5 | <0.5 | <0.5 | <0.8 |
| Dynamic range | | | 3–350 K | | |
| CAL precision | | | 1 K (180~320 K) | | |

To more quickly obtain the dynamic environmental information of the majority of the global oceans, the HY-2C and HY-2D were designed with non-sun-synchronous inclined orbits, and only carry the payloads of radar altimeter, microwave scatterometer, calibration microwave radiometer (CMR), DCS, and AIS. The design about the main sensors on the HY-2C/D satellites is the same as that of HY-2A/B. The HY-2C and HY-2D satellites were

successfully launched on 21 September 2020 and 19 May 2021, respectively [http://www.nsoas.org.cn]. At the present time, China has constructed the first batch of its marine dynamic environmental satellite constellations. The joint observations of three satellites were adopted. The observational efficiency of the sea surface wind fields is detailed in Figure 1. The sea surface wind field observation range from the HY-2 series satellites reaches nearly 70% of the global sea area in a three-hour period; 86% in six hours; and 95% in 12 h. It is considered that if combined with the Meteorological operational satellite (MetOp) satellite, the wind field observational effectiveness will be further improved. Coupled with the OceanSat-3 equipped with scatterometer to be launched by (Indian Space Research Organisation) ISRO in 2022, the wind field observation efficiency will be further improved.

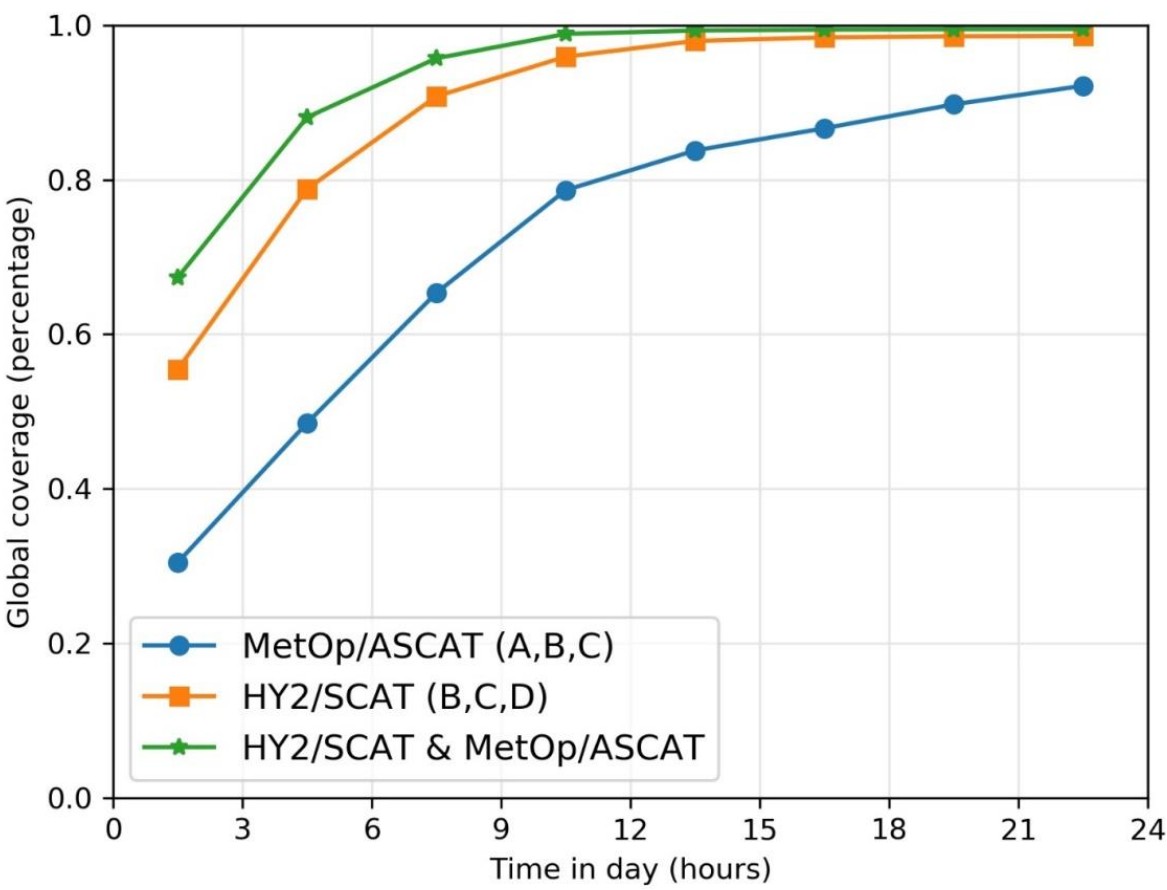

**Figure 1.** Global sea-area coverage of the dynamic satellite constellation scatterometer.

The FY-3E satellite, launched by the National Satellite Meteorological Center (NSMC) of the China Meteorological Administration on 5 July 2021, carries the world's first dual frequency and dual polarization conical scanning radar, which obtains high-precision wind field measurements through on-board internal calibration and on orbit active external calibration [30]. At present, the satellite is in orbit undergoing testing. FY-3E wind radar works in C and Ku bands at the same time, which can ensure the high-precision measurement of low and medium wind speed field, but also measure the high wind speed wind field of typhoon level. The sea surface wind field data obtained by FY-3E wind radar is worth looking forward to.

Similarly, the observation efficiency of radar altimeter using the HY-2 constellation is analysed in this paper. Figure 2 details the maximum radius of at least one observation point in a circle if we randomly find a point on the earth and draw a circle around the point. The blue and orange curves in the figure show that the coverage of China's three altimetry satellites is slightly lower than that of the four international altimetry satellites.

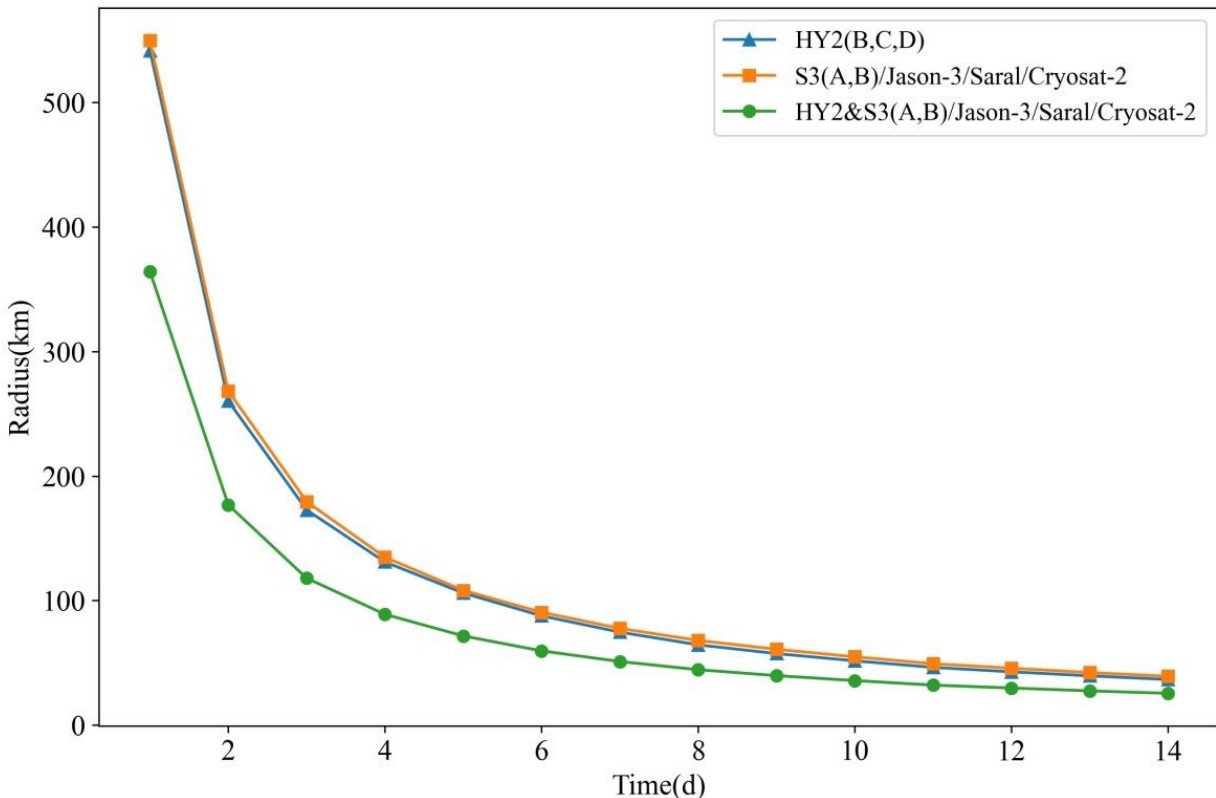

**Figure 2.** Observation efficiency of radar altimeter constellation.

### 2.2. *Progress History of China's Marine Surveillance Satellites*

The GF-3 satellite in China's high-resolution Earth observation system is China's first marine surveillance satellite. On 10 August 2016, China successfully launched the GF-3 satellite with the Long March-4C rocket at the Taiyuan Satellite Launch Center. The GF-3 satellite is China's first C-band multi-polarized synthetic aperture radar (SAR) satellite with a spatial resolution of 1 m [10]. The missions and purposes of the GF-3 satellite are different when compared with the previously developed satellites. For example, a new system and multi-polarization design were adopted for the purpose of collecting as much information as possible and transmit the data to the ground, thereby providing a basis for obtaining backscatter with four polarizations (VV, VH, HH and HV) from the Earth's surface. The GF-3 satellite is currently the highest-resolution C-band and multi-polarized satellite in the world. Moreover, the microwave images acquired by the GF-3 satellite have high performance and can be used for not only geometric information of specific targets, but also quantitative inversion processes.

The GF-3 satellite has 12 imaging modes (see Figure 3) which include a traditional strip imaging mode and a scanning imaging mode, as well as a wave imaging mode and a global observation imaging mode for marine applications. At the present time, it is the SAR satellite with the most imaging modes in the world. It has been found that with its wide imaging width and high spatial resolution, it can provide large-scale surveying processes as well as survey specific areas in detail [31]. Therefore, the GF-3 satellite has been able to meet the imaging needs of different users for different targets.

In addition, the GF-3 satellite has significantly improved the remote sensing earth observational abilities. It is expected that the reliable and stable high-resolution microwave image data provided by the GF-3 satellite will effectively supply the needs of the users for high-resolution civil microwave remote sensing satellite data. At the same time, the GF-3 satellite has successfully obtained high-resolution multi polarization or full polarization microwave images [32], providing high-quality and high-precision earth observation data for users in global marine scientific research, land change monitoring, water conservancy

applications and other fields. It is widely used in marine fields, such as marine disaster, oil spill, green tide, marine wave inversion, marine target ship, sea ice identification and so on [33–35].

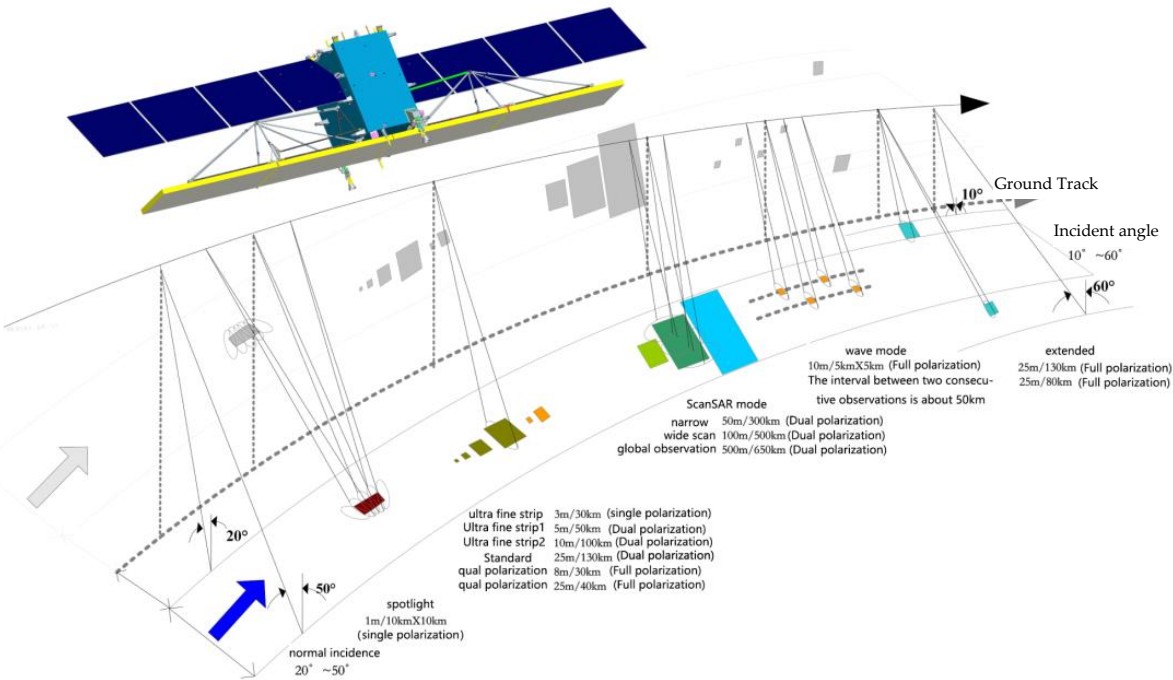

**Figure 3.** Observation diagram of the GF-3 satellite. The grey rectangle represents the schematic of different modes.

## 3. Progress History of China's Marine Microwave Satellite Remote Sensing Technology

With the successful application of the SZ-4 [36] multi-mode microwave remote sensor, China's marine microwave (dynamic environments) remote sensing has been gradually developed. However, most of the past development has focused on the introduction and learning of the existing global algorithm models and independent innovation has been lacking. Since China launched the HY-2A marine dynamic environment satellite on 16 August 2011, marine microwave remote sensing technology has been rapidly developing, and many innovative results have been achieved in the development and applications of algorithms. The successful launch of the HY-2B satellite on 25 October 2018 marked the transition of China's marine dynamic environment satellites from research to operations. The marine dynamic environmental elements now mainly include sea surface wind fields, waves, oceanic currents, temperatures, salinity, and others. At the present time, data regarding all the aforementioned elements can be obtained through remote sensing, and the accuracy has been gradually improving.

### 3.1. Retrieval Technology for Marine Dynamic Environmental Elements

Regarding the retrieval algorithms for significant wave heights, a new algorithm with skewness coefficient was derived based on a second-order theoretical echo model [37]. Then, singular value decomposition (SVD) filtering was introduced, and multiple re-tracking schemes were successfully obtained according to the different retrieval parameters of the maximum likelihood estimation algorithm. Comparative analysis was used to determine which four-parameter model was the most suitable for the HY-2A satellite's radar altimeter waveform retrievals, and the significant wave height retrieval accuracy reached 0.31 m [9]. Jiang et al. [38] re-evaluated the significant wave height products obtained by the HY-2A satellite's radar altimeter using the retrieval of the four-parameter re-tracking algorithm. It was found that the accuracy of the product was significantly improved when compared

with the previously used three-parameter re-tracking algorithm. It has been confirmed that sea state bias is a very important error source in the current sea level measurements obtained by radar altimeters [39]. Chinese and international researchers have proposed many correction algorithms for sea state bias [40–43]. However, at the present time, all the effective correction algorithms are empirical models [44]. A neural network-based sea state correction algorithm has been developed based on those empirical models in recent years [45].

The sea surface wind fields obtained by satellite remote sensing include two key elements: speed and direction. Microwave scatterometers, microwave radiometers, radar altimeters, and synthetic aperture radar all can obtain sea surface wind speeds. However, microwave radiometers and radar altimeters cannot observe wind directions due to the limitations of their observational principles. Therefore, the observational accuracy of each payload in different wind speed ranges will tend to vary [46]. Regarding microwave scatterometer sea surface wind field retrieval algorithms, multi-solution retrieval algorithms and two-dimensional variation fuzzy solution removal methods are generally used to process the HY-2A satellite's scatterometer data [47]. The results have shown that the accuracy of the wind speeds and directions were greatly improved [48,49]. In addition, the removal abilities of the wind direction fuzzy solutions were found to be obviously enhanced. In particular, for the wind field retrieval results under typhoon conditions, the accuracy of the wind direction retrievals has been significantly improved. Wang [50] applied a multiple solution scheme (MSS) and the two-dimensional variation analysis method (2DVAR) proposed by the Royal Netherlands Meteorological Institute for wind field retrievals from the HY-2A satellite's microwave scatterometer and achieved satisfactory results. In terms of radar altimeter sea surface wind speed retrievals, Wang et al. [51] established the relationship between the Automatic Gain Control (AGC) from the HY-2A satellites radar altimeter and the sea surface wind speeds and solved the low wind speed problem of the HY-2A satellite's radar altimeter sea surface wind speed retrievals in their breakthrough research results.

In the research of sea surface temperature retrieval algorithms, Wang et al. [52] established a retrieval algorithm for sea surface temperatures, sea surface wind speeds, atmospheric water vapor content, and other marine environmental parameters using radiation transfer equation simulations. In addition, Sun et al. [53] developed a grid-based method of sea surface temperature data fusion by applying optimal interpolation technology and expanded the products of the sea surface temperature data obtained via satellite remote sensing.

In addition, in research involving sea ice concentration retrieval algorithms, Shi et al. [54] conducted statistical analysis regarding the spectral gradient rates and polarization gradient rates of typical sea areas and determined the brightness temperature eigenvalue required for calculating sea ice concentrations. The calculation results were found to be consistent with the two state-of-the-art sea ice concentration products provided by the National Snow and Ice Data Center and the University of Bremen (Germany) [54]. Henan University of Technology [55] invented an algorithm that was subsequently patented: an improved ASI (the Arctic Sea Ice) sea ice concentration retrieval algorithm. The obtained sea ice concentration values were found to be significantly lower than the ASI algorithm results, with the sea ice area reduced by approximately 15%. The number of pixels with ice concentrations above 0.15 was approximately 28.6% lower than the obtained using the traditional ASI version 5.3 [56]. The results of the improved ASI algorithm are very different from those of the traditional ASI algorithm [56], which is closer to the results of NASA team [57].

The retrieval algorithm of marine environmental parameters designed according to the characteristics of China Marine microwave remote sensing satellite plays a very important role in the data processing system.

### 3.2. Calibration Technology for Marine Dynamic Environmental Satellites

The calibrations of satellite remote sensors are key to the quantitative applications of data. In the research regarding calibration methods, Zhang and Lin [58] summarized the distributions of calibration fields, as well as calibration methods for the backscattering coefficient and sea surface heights of satellite radar altimeters. In the Yellow Sea, the measured tide gauge data were extrapolated into sub-satellite points to calibrate the sea surface heights, thereby providing a new method for the on-orbit calibrations of radar altimeters. Yan et al. [59] analysed the factors affecting the accuracy of GPS buoy height measurements, and accumulated experience for the use of GPS buoys for the absolute calibrations of sea surface heights. Yang et al. [60] summarized the main technologies and developmental trends of satellite altimeter calibrations in China and internationally and put forward suggestions on the construction and planning of China's calibration fields. After summarizing the methods and content of satellite radar altimeter calibrations and tracking and investigating the configurations of international altimeter calibration fields, Jiang et al. [14] put forward a basic idea for the construction of China's marine satellite altimeter calibration field. In addition, the geophysical parameters and hydro meteorological parameters affecting altimeter calibrations in sea areas around the calibration fields were preliminarily analysed, and a basic scheme for the configuration of China's calibration field was successfully determined. As a result, the foundation for the construction of China's satellite radar altimeter calibration field was laid out.

### 3.3. Precise Orbital Determination Technology

For the precise orbital determinations of marine dynamic environment satellites, a simplified dynamic method is generally used to estimate the parameters through non-differential state estimations and non-differential ambiguity fixation [61,62]. It has been found that the accuracy of the orbital determinations and positioning results can be improved using such a method [63]. In terms of DORIS and laser precise orbital determinations, a dynamic method is usually adopted to achieve the precise orbit determinations of medium orbit ephemeris (MOE) and precision orbit ephemeris (POE) satellites. The applications of the aforementioned orbital determination methods have resulted in the precise orbital determinations of China's marine dynamic environment satellites reaching the international advanced level [64].

Zhu et al. [65] examined an inter-epoch differential processing method that differed from the traditional phase observation data processing methods. The phase observational data were converted into distance change rate observational data, and the relative errors were corrected. Then, a perturbation model of the macro-model and empirical force from the HY-2 satellite was established. In addition, based on the dynamic orbital determination principle, precise orbital determinations were obtained using DORIS phase observational data. Orbital determinations were successfully carried out using the DORIS measured data from the HY-2 satellite. The preliminary calculation results showed that the radial orbital errors were smaller than 2 cm and the three-dimensional position errors were approximately 10 cm [66]. Jiang et al. [16] summarized the DORIS calibration technology, which is one of the main orbital determination technologies of the HY-2A satellite, and verified the accuracy of the DORIS orbital determinations using satellite laser ranging (SLR) verifications and independent orbital comparisons. The results confirmed that the DORIS precise orbital determination technology could be applied to the HY-2 series satellites. Lin et al. [64] established dual-frequency GPS precise orbital determination technology using the dual-frequency GPS carried by the HY-2 satellites. After verifications were completed, it was determined that the orbital determination accuracy using the dual-frequency GPS could meet the requirements. Subsequently, the HY-2B satellite system was not equipped with a DORIS orbital determination system, and only a dual-frequency GPS was used for operational orbit determinations. This had laid a foundation for the subsequent use of the Beidou System in China for precise orbital determinations.

Based on the precise orbit determination technology of the HY-2 satellites, Fan et al. [66] proposed a real-time precise orbital determination method constrained by ultra-fast precise ephemeris, which was aimed at the fact that the real-time filtering precise orbital determinations of the Beidou System satellite with broadcast ephemeris as the starting orbit often require a long convergence time. Next, based on the measured data of 51 stations in the MGEX (Multi-GNSS Experiment) tracking network for seven consecutive days, the real-time precise orbit of the Beidou System satellite was determined using a square root information filtering method. The real-time filtering orbital accuracy of the Beidou System satellite was evaluated with reference to a three-day post solution orbit. The current maximum accuracy is about 5 cm.

## 4. History of Progress in China's Marine Surveillance Satellite Remote Sensing Technology

The GF-3 satellite is China's marine surveillance satellite and is equipped with 1 m resolution synthetic aperture radar (SAR). It is China's first C-band multi-polarized SAR imaging satellite with that high resolution and plays an increasingly important role in the fields of oceanic theory and applications. For SAR marine remote sensing, the sea surface roughness is the main factor affecting radar backscattering under certain radar parameters and orbital conditions [67–69]. The sea surface roughness measured by SAR is caused by surface tension waves and short gravity waves ranging from several centimetres to tens of centimetres. The imaging ability of SAR for oceanographic features or phenomena (such as winds, currents, waves, fronts, oil film, vortexes, internal waves, and underwater topography) is dependent on the degree to which those features, or phenomena, change the sea surface roughness in various ways.

### 4.1. Oceanographic Information Retrieval Technology

From the perspective of wave information extractions, Xu et al. [70] developed an algorithm, which included a wave parameter retrieval method based on co-polarized SAR data. Based on the acquisition of the full-polarization backscattering coefficient of the sea surface, the wave parameters are calculated. Regarding sea surface wind field information extraction processes, Zhang [71] introduced the polarizability model XPR2, that included both radar incidence angles and sea surface wind speeds. The XPR2 model was fitted with 56 dual-polarized TS-X and TD-X images and matched ECMWF wind field data. The validity of the XPR2 model was verified by 38 HH polarized Terra SAR-X images and corresponding buoy measurements. The verification results showed that the root mean square error of the XPR2 model retrieval results was 1.79 m/s, and the deviation was 0.68 m/s. Lin et al. [72] proposed typhoon identification technology in combination with HY-2A scatterometer and GF-3 C-band SAR wind fields and obtained fine typhoon observations using China's independently developed marine satellite.

In another related study, Song et al. [73] adopted the advanced synthetic aperture radar large-amplitude model data of ESA (European Space Agency) ENVISAT, removed the frequency shifts caused by the relative motions of the Earth and the satellite based on the theoretical model of Doppler frequency shifts, and eliminated the influencing effects of the sea surface wind fields and Bragg scattering using the C-band empirical model CDOP (C-Band Doppler Frequency) and a dual-scale convergent scattering model, respectively, thereby obtaining the retrieval of high-resolution sea surface flow fields.

Zou et al. [74,75] carried out analyses of oil spill monitoring parameters from the aspects of bands, polarization modes, incidence angles, and so on, using SAR data based on the calculations of backscattering coefficients. In the this study, the weights of each type of index were obtained by an analytic hierarchy process (AHP) in combination with such main indexes as oil spill shape parameters, texture characteristic indexes, and physical characteristic indexes. An oil spill remote sensing monitoring method was used to analyse the data, and a multi-index oil spill information extraction method based on SAR data was proposed.

*4.2. Marine Target Recognition Technology*

Since the early 1990s, in accordance with the research results of various international research institutions and colleges regarding SAR target detection and recognition methods combined with airborne SAR data, China's researchers have carried out studies regarding the detection and recognition of large strategic targets such as airports, ports, oil depots, and thermal power stations [76,77].

In terms of marine target monitoring, the current mainstream SAR ship targeting detection methods can be divided into the following two categories according to the most important grey features in SAR images: (1). Ship target detections based on background clutter statistical distributions; and (2). Ship target detections based on polarimetric decomposition. Leng et al. [78] determined the spatial distributions of targets using a kernel density estimator, and successfully eliminated the false detection results of targets caused by azimuth ambiguities and the scattering intensities of clutter pixels similar to ships. It was found that when other parameters remained unchanged, the systems with longer wavelengths had a lower probability of ambiguity in the oblique direction. The reason for those results was found to be related to the fact that the ambiguity had exceeded the target area in the oblique direction. Therefore, in the SAR image ship detections, Wei et al. [79] focused on the removal of azimuth ambiguity and proposed corresponding solutions. For example, according to the characteristics of the high-resolution SAR data, Wang et al. [80] calculated the three characteristics of the kernel density estimations (KDE), length-width ratios, and pixel numbers, and then determined the weight proportion of the characteristics using an AHP. Subsequently, the confidence degree of the preliminarily detected target belonging to the ship was obtained through the optimization of multiple characteristics for the discrimination threshold of the ship characteristics, finally achieving the optimized ship detection results. Liu [81] proposed a signal-to-noise ratio enhancement method. It was considered that such a method would significantly improve the signal-to-noise ratio of the SAR images. Then, based on the that method, small targets on the sea surface were monitored, and strong results were obtained.

Li [82] detected ships using SAR image data, analysed the characteristics of the ships, and then determined the detailed information of various ships, extracted ship target slices, and established a test sample library based on the matching information provided by AIS (automatic identification system). The extracted ship features were then used for classification and recognition of ship types. Using deep learning technology, ship classification in high resolution SAR has also made some progress. The accuracy of classification is more than 95% [83].

## 5. Development Trends of Marine Microwave Remote Sensing Satellites in China

*5.1. Development Trends of Marine Dynamic Environment Satellites in China*

In the future, China will continue to strengthen the monitoring of marine dynamic environments and provide monitoring information regarding sea surface temperatures, waves, sea surface heights, and other factors used in the forecasting of global climate changes and marine disasters. China has planned two batches of marine dynamic environment satellite constellations. The first constellation, which includes the HY-2B, HY-2C, and HY-2D satellites, was established on 19 May 2021. The satellites of this constellation will be replaced by the HY-2E, HY-2F, and HY-2G satellites at the end of their respective lifespans in order the establish China's second constellation. The two constellations will continuously monitor the global marine dynamic environment information in a comprehensive way [16,17].

In addition to building marine dynamic environment satellite constellations based on existing mature technologies, China is also exploring new microwave remote sensing payloads for marine dynamic environments. These include wide-swath interferometric radar altimeters, dual-frequency full-polarized microwave scatterometers, and full-polarized microwave radiometers. As follow-on satellites of the HY-2 series satellites, the main purpose is to improve the sea surface height values obtained via sub-satellite point measurements

to wide-swath measurements [84]. The maximum wind speed measurement capacity is increased from 24 m/s to 50 m/s, and the sea surface temperature measurement accuracy can reach higher than 0.5 °C based on the existing observational accuracy [84].

Furthermore, China is exploring new technology for observations of sea surface salinity levels and surface flow fields, using microwave remote sensing technology [85].

All China's ocean satellite data are free for users all over the world (The data download address is https://osdds.nsoas.org.cn/). To facilitate the use of users related to global marine science, the data product format design of China ocean satellite is basically the same as that of ocean observation satellites of other countries. Therefore, China's ocean satellites can form a larger virtual constellation with the ocean satellites of other countries to serve global climate change studies, disaster prevention and control, etc.

*5.2. Development Trends of Surveillance Satellites in China*

Similar to the marine dynamic satellite constellation, China is also building a monitoring satellite constellation, which is expected to be completed in 2022 [86]. The main payload of that constellation is a 1 m resolution synthetic aperture radar. The constellation will be composed of two satellites. After completion, it will provide high-effectiveness and high-resolution observational data for global marine disaster prevention and mitigation, as well as maritime search and rescue.

China's surveillance satellite constellation will not only play an important role in ocean applications, but also provide quasi real-time monitoring for such aspects as agriculture, forest coverage, and changes to grassland regions.

**6. Conclusions and Prospects**

Since the 1960s, through the continuous efforts of several generations of marine remote sensing researchers, China has achieved fruitful results in marine remote sensing technology [12,13,16]. China has formulated a long-term plan for the development of marine satellites and has developed three series of marine satellites: Oceanic colour; oceanic dynamic environments; and oceanic monitoring. These three series will gradually form China's marine space monitoring network. Since the launch of China's first marine satellite HY-1A in 2002, China has launched four ocean-colour satellites, five marine dynamic environment satellites (including China-France Oceanography Satellite (CFOSAT)), and one marine surveillance satellite, thereby obtaining the daily observations of the global marine environment. As a result, the quality of the remote sensed products has been greatly enhanced. Satellite products play important roles in marine resource and environment monitoring processes, marine disaster prevention and mitigation processes, and maritime security management in China.

With the development of marine satellites and the demand for satellite data processing, China has successfully made important breakthroughs in marine satellite remote sensing retrieval technology [87,88]. From the aspect of marine dynamic environment satellite remote sensing technology, extraction algorithms for significant wave heights and sea surface wind fields for China's independent marine microwave (marine dynamic environment) satellite were proposed in combination with the design characteristics of China's marine microwave (marine dynamic environment) satellite, and then applied to the operational system. It was confirmed that the accuracy of the data products had successfully reached the international leading level. Considering that there is no calibration field for marine microwave (marine dynamic environment) satellites in China, a calibration method using GPS buoys and tide gauges was put forward. It was observed that for precise orbital determinations, the parameters could be estimated by a simplified dynamic method through non-differential state estimations and non-differential ambiguity fixation for the purpose of improving the accuracy of orbital determinations and positioning. In terms of marine surveillance satellite remote sensing technology, breakthroughs have been made in the traditional extractions of marine environment elements, and major progress has been achieved in marine target recognition.

It is expected that through the efforts made during the next 5 to 10 years, China's marine satellite remote sensing technology will play an important role in the world [89]. First, China's marine dynamic environment satellites will gradually form a network that will provide simultaneous in-orbit operation of one polar orbit and two inclined orbit marine dynamic environmental satellites. This will meet the needs of monitoring mesoscale and subspecies scale marine dynamic phenomena, and successfully complete the world's first marine dynamic environment constellation. Second, China is vigorously carrying out research regarding new marine microwave satellite remote sensing detection technology and is expected to make important breakthroughs in the fields of wide-swath radar altimetry imaging, salinity remote sensing, and synthetic aperture radar, thereby providing key technical support for China's development of new marine satellites [90]. The accurate observations of sea surface temperatures, sea surface salinity, sea surface wind fields, sea waves, surface flow fields, and other marine dynamic environment elements will provide effective scientific data for early warnings and the forecasting of global climate changes, air-sea interactions, and catastrophic sea states.

**Author Contributions:** Conceptualization, M.L. and Y.J.; methodology, Y.J.; validation, M.L. and Y.J.; investigation, M.L.; resources, Y.J.; data curation, Y.J.; writing—original draft preparation, Y.J.; writing—review and editing, M.L.; funding acquisition, M.L. All authors have read and agreed to the published version of the manuscript.

**Funding:** This research was funded by National Natural Science Foundation (42192561, 42192531), Key Special Project for Introduced Talents Team of Southern Marine Science and Engineering Guangdong Laboratory (Guangzhou) (GML2019ZD0302).

**Institutional Review Board Statement:** Not applicable.

**Informed Consent Statement:** Not applicable.

**Data Availability Statement:** Not applicable.

**Acknowledgments:** The Jason-3 GDR-D data were obtained from AVISO. Sentinel-3 data were obtained from ESA. The HY-2B/C/D and GF-3 data were obtained from https://osdds.nsoas.org.cn/#/, (18 December 2021). The MetOp/ASCAT data were obtained from ftp://ftppro.knmi.nl/scat/netcdf//#/, (7 January 2022). In addition, the authors would like to thank Zhixiong Wang for his assistance in the data processing.

**Conflicts of Interest:** The authors declare no conflict of interest.

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
