# Peer review of "Past, Present and Future Marine Microwave Satellite Missions in China"

_remotesensing, doi:10.3390/rs14061330_

Round 1
Reviewer 1 Report
The manuscript provides an overview of China's passive- and active-microwave environmental satellites, and a summary of results therefrom. It will be of interest to readers of the special issue and it fits the topic.
There is some repetition, but perhaps that can be excused in a review paper. English usage will be acceptable with a few revisions as noted below.
1. The term used in English is scatterometer, not "scatter-o-meter" (numerous places in the text).
2. Table 1 appears to have a formatting error.
3. The caption of Figure 3 should explain what the gray rectangles signify.
4. Lines 336 and 338: "cold-air reflectors" probably means cold-space reflectors.
5. Line 438: AHP should be defined here instead of at line 465.
6. Line 497: the meaning of "Sea surface wind speed measurements range from 2 to 24 m/s to 2 to 50 m/s" is not clear.
Author Response
Thanks for your comments on our paper. We have revised the paper according to your comments. We have deleted some repetitive presentation, and some grammar and spelling errors have also been corrected. We have made corresponding revision according to your advice. Words in blue are the changes we have made in the text. The following is the answers and revisions I have made in response to the reviewers' questions and suggestions on an item by item basis.
Comment 1: The term used in English is scatterometer, not "scatter-o-meter" (numerous places in the text).
Response: Replace "scatter-o-meter" with "scatterometer" in the text.
Comment 2:Table 1 appears to have a formatting error.
Response: The wrong format of Table 1 Table 2 and Table 3 has been modified.
Comment 3: The caption of Figure 3 should explain what the gray rectangles signify.
Response:
Comment 4:Lines 336 and 338: "cold-air reflectors" probably means cold-space reflectors.
Response: Replace "cold-air reflectors" with "cold-space reflectors" in the text.
Comment 5:Line 438: AHP should be defined here instead of at line 465.
Response: The AHP is defined on line 438.
Comment 6:Line 497: the meaning of "Sea surface wind speed measurements range from 2 to 24 m/s to 2 to 50 m/s" is not clear.
Response: The maximum wind speed measurement capacity is increased from 24m / s to 50m / s. It is modified in this text.

Reviewer 2 Report
This paper presents a review of microwave remote sensing activities in China, including the development of satellite technology, processing algorithms, and future plans. The description of satellite technology development activities and plans is cogent and presumably authoritative. The development of data processing algorithms and information extraction methods is as important as the development of sensor technology, and this development is discussed in section 3 of this paper. However, this area of research tends to be somewhat more abstract and difficult to explain in general or non-specialist terms. The difficulty of summarizing the development of data processing methods may be compounded by language differences or translation issues. In any case, this section of the paper needs to be greatly revised or dropped from the paper. I would suggest attempting to describe the general problem that each research project or area is intended to address, and to summarize in the simplest possible terms the means or approach being taken to solve those problems. A few examples of what I consider to be bad writing are as follows:
Lines 257-258: "a second-order theoretical echo model ... was derived based on a second-order theoretical echo model." This doesn't make sense. Maybe replace the first phrase with "a new algorithm"?
Line 308: the phrase "invented an algorithm patent" is not correct. Patents are not invented, but they may be awarded for innovative algorithms.
Line 308: The abbreviation "ASI" should be defined.
Line 309: what is "retrieval algorithm manufacturing technology"? This phrase doesn't make any sense to me.
Line 313: what is the evidence that the lower results are "significantly improved", i.e. more accurate?
Line 314: what does "effectively reduced the impacts of weather on the high-frequency data" mean?
Lines 332-334: I don't think this is an accurate summary of Zhou et al's research. Might be a language translation issue.
Line 357: what is "macro surface force"?
Line 398: "invented an algorithm patent" again – should be "patented an algorithm" or "invented an algorithm that was subsequently patented"
Lines 398-412: this discussion should be shortened and clarified.
Lines 447 -448: what is the difference between "detections based on polarimetric decomposition" and "detections based on polarization features", i.e. categories 2 and 3?
Although the description of satellite technology is on the whole quite good, here are a few suggestions to clarify the discussion or to modify the terminology to conform with conventional usage:
Line 88: "study" should be changed to "review"
Sixteen occurrences, starting on line 109, of "scatter-o-meter" should be changed to "scatterometer" (not hyphenated)
Line 115: "et al" should be "etc."
Line 121: sentence is unclear, and "up to 16 antennas" is not appropriate. Presumably the number of antennas is known, and the statement should be definite.
Table 2: is the stated pulse width of 1.5 ms the actual transmitted length or the effective length resulting from pulse compression? Also, regarding the wind direction accuracy, an ESA report (https://earth.esa.int/web/eoportal/satellite-missions/h/hy-2a) states that the wind direction retrieval of the HY-2 scatterometer "still has some problems which need to be investigated". Do the authors wish to comment about that?
Figure 2: this figure could be described a little more clearly. I think that what the figure means is that relative to any given point on the earth's surface a measurement will be made within the distance shown on the vertical axis during the time interval shown on the horizontal axis. The last sentence above this figure is not very helpful because it is not consistent with the plot.
Line 213: "space" should be "spatial"
Line 523: "majorly improved" is not good grammar
Author Response
Thank you very much for your valuable comments on the revision of the manuscript.
Comment 1: Lines 257-258: "a second-order theoretical echo model ... was derived based on a second-order theoretical echo model." This doesn't make sense. Maybe replace the first phrase with "a new algorithm"?
Response: According the reviewer’s comment, we have corrected the sentence.
Comment 2: Line 308: the phrase "invented an algorithm patent" is not correct. Patents are not invented, but they may be awarded for innovative algorithms.
Response: According the reviewer’s comment, we have corrected the sentence. Furthermore, we have had the manuscript polished with a professional assistance in writing.
Comment 3: Line 308: The abbreviation "ASI" should be defined.
Response: Thank you for your careful work. We have added the definition of ASI.
Comment 4: Line 309: what is "retrieval algorithm manufacturing technology"? This phrase doesn't make any sense to me.
Response: We are so sorry for this wrong phrase. We have revised the sentence.
Comment 5: Line 313: what is the evidence that the lower results are "significantly improved", i.e. more accurate?
Response: Thank you very much for pointing out the exact use of adverbs. We have replaced “significantly” with “observably”.
Comment 6: Line 314: what does "effectively reduced the impacts of weather on the high-frequency data" mean?
Response: We wanted to express the high-frequency changes of sea ice concentrations caused by weather changes in a short time. So we have replaced “high frequency data” with “high frequency variation”.
Comment 7: Lines 332-334: I don't think this is an accurate summary of Zhou et al's research. Might be a language translation issue.
Response: Thank you very much for your careful work. In fact, Zhou et al. only studied the cold space of HY-2A microwave radiometer.
Comment 8: Line 357: what is "macro surface force"?
Response: Thank you very much. This is a language translation issue. We have carefully consulted other literature. This should be macro model. A macro-model was developed for modeling radiation pressure forces in HY-2 precision orbit determination.
Comment 9: Line 398: "invented an algorithm patent" again – should be "patented an algorithm" or "invented an algorithm that was subsequently patented"
Response: Thank you for your suggestions. According the reviewer’s comment, we have corrected the sentence.
Comment 10: Lines 398-412: this discussion should be shortened and clarified.
Response: Thank you for your suggestions. This part of the discussion is simplified.
Comment 11: Lines 447 -448: what is the difference between "detections based on polarimetric decomposition" and "detections based on polarization features", i.e. categories 2 and 3?
Response: There is no substantive difference between the two statements. According the reviewer’s comment, we have corrected the sentence.
Although the description of satellite technology is on the whole quite good, here are a few suggestions to clarify the discussion or to modify the terminology to conform with conventional usage:
Comment 12: Line 88: "study" should be changed to "review"
Response: According the reviewer’s comment, we have corrected the statement.
Comment 13: Sixteen occurrences, starting on line 109, of "scatter-o-meter" should be changed to "scatterometer" (not hyphenated)
Response: According the reviewer’s comment, we have corrected the sentence.
Comment 14: Line 115: "et al" should be "etc."
Response: According the reviewer’s comment, we have corrected the sentence.
Comment 15: Line 121: sentence is unclear, and "up to 16 antennas" is not appropriate. Presumably the number of antennas is known, and the statement should be definite.
Response: According the reviewer’s comment, we have removed “up to”.
Comment 16: Table 2: is the stated pulse width of 1.5 ms the actual transmitted length or the effective length resulting from pulse compression? Also, regarding the wind direction accuracy, an ESA report (https://earth.esa.int/web/eoportal/satellite-missions/h/hy-2a) states that the wind direction retrieval of the HY-2 scatterometer "still has some problems which need to be investigated". Do the authors wish to comment about that?
Response: It is the actual transmitted length.
Please note that, the HY-2B, HY-2C, and HY-2D satellites are operating now, and their backscatter measurements or wind products are highly consistent and of good quality. Please refer to these peer-reviewed publications:
[1] Zhixiong Wang, Juhong Zou, Youguang Zhang, Ad Stoffelen, Wenming Lin, Yijun He, Qian Feng, Yi Zhang, Bo Mu, and Mingsen Lin. 2021. "Intercalibration of Backscatter Measurements among Ku-Band Scatterometers Onboard the Chinese HY-2 Satellite Constellation" Remote Sensing 13, no. 23: 4783. https://doi.org/10.3390/rs13234783.
[2] Zhixiong Wang; Ad Stoffelen; Juhong Zou; Wenming Lin; Anton Verhoef; Yi Zhang; Yijun He; Mingsen Lin, "Validation of New Sea Surface Wind Products From Scatterometers Onboard the HY-2B and MetOp-C Satellites," in IEEE Transactions on Geoscience and Remote Sensing, vol. 58, no. 6, pp. 4387-4394, June 2020, doi: 10.1109/TGRS.2019.2963690.
[3] Zhixiong Wang; Juhong Zou; Ad Stoffelen; Wenming Lin; Anton Verhoef; Xiuzhong Li; Yijun He; Youguang Zhang; Mingsen Lin . "Scatterometer Sea Surface Wind Product Validation for HY-2C," in IEEE Journal of Selected Topics in Applied Earth Observations and Remote Sensing, vol. 14, pp. 6156-6164, 2021, doi: 10.1109/JSTARS.2021.3087742.
Or the reviewer could refer to KNMI website for HY-2B and HY-2C winds validation reports: https://scatterometer.knmi.nl/hy2b_25_prod/
However, the HY-2A data are indeed degraded in years after 2015.
Comment 16: Figure 2: this figure could be described a little more clearly. I think that what the figure means is that relative to any given point on the earth's surface a measurement will be made within the distance shown on the vertical axis during the time interval shown on the horizontal axis. The last sentence above this figure is not very helpful because it is not consistent with the plot.
Response: What this figure wants to express is that if we find a random point on the earth's surface and draw a circle with this point as the center, there must be at least one satellite altimetry observation within the radius of the vertical axis within the time represented by the horizontal axis.
Comment 17: Line 213: "space" should be "spatial"
Response: Thank you for your suggestions.
Comment 18: Line 523: "majorly improved" is not good grammar
Response: Thank you for your suggestions.

Reviewer 3 Report
The paper describes the history and future projects of Ocean microwave remote sensing missions. Which is very interesting and informative.
However, it's missing a huge amount of citation. As a general rule, if you present a satellite, an instrument, a group, institute, mission, dataset, etc you need to present a reference. Also, if you make claims of something improving the accuracy of retrievals or developing a method, you need to include references.
The introduction specially should be a bibliographical review, and in this case it does not contain any citations.
I believe that the lack of references is the main issue of the paper, but easy to fix.
The figures need to be better explained in text and in the captions. What do the colors represent?
Some English editing is necessary. Throughout the manuscript, the authors used the word realizing, which sounds very estrange when reading. Considering changing for providing, obtaining, developing, etc. Or rephrasing the sentence to make it more clear.
Finally, the authors claim that the datasets are free for users worldwide, however, no reference has been made as to how to access this data or where to find it. I suggest adding this, so readers can see where it is.

Author Response
Thank you very much for your valuable comments on the revision of the manuscript. According to your revision comments, a large number of references have been added. At the same time, according to your suggestions on the revision of the manuscript, it has been revised one by one. The revised manuscript is attached.

Round 2
Reviewer 2 Report
I am recommending publication after minor revisions, but I do think the manuscript needs improvement, and could be an embarrassment if published in its present form, so I would encourage you to consider the following comments.
The authors have incorporated the simple suggestions I made regarding language and clarity but have ignored the more substantive comments. For example, in the discussion of the ASI algorithm on lines 321-329, I commented that obtaining different results does not necessarily imply that the new results are better. Their response was to add the word "obviously" to a conclusion that is not by any means obvious. Where is the evidence that the new results are more accurate? If they are not prepared to defend this conclusion, the authors should simply state the difference without claiming an improvement. Also, how does the new algorithm reduce the impacts of weather? (Not to mention that the impacts are presumably on the measurement, not on the phenomenon being measured.) By the way, I think that ASI stands for "Arctic Sea Ice" generally, but there are specific ASI algorithms, such as the one developed during the ARTIST project, and the specific algorithm should be referenced when making comparisons. For example, in the first use of the term, ASI should be defined parenthetically as simply (Arctic Sea Ice) but in subsequent references, when it is stated that the results of the new algorithm are "significantly lower than the ASI algorithm" the specific version should be specified, such as version 5.6i for example. Similarly, "the traditional ASI algorithm" should be replaced by a reference to the specific version.
The discussion on lines 345-348 commenting on the results of Zhou et al [59] were very slightly modified but not at all clarified. The authors should refrain from adding details that are not understandable in the context of a general summary. I would suggest something like "Zhou et al [59] developed an improved calibration procedure based on cold-space radiation measurements." This comment applies to the other summaries as well, which in general incorporate too much specialized terminology, some of it incorrectly.
In reference to Figure 2, the authors have not removed the sentence "It can be found that in the 10 day observation period of HY-2C/D, the minimum value of the maximum radius is 86 km" despite the fact that this is inconsistent with the Figure, as I pointed out. They added the sentence "In the context of global warming, satellite altimetry data all over the world are basically free for scientists to use" which seems meretricious and irrelevant except perhaps to explain the inclusion of international data in the figure. The point of the figure seems to be that the Chinese data is basically equivalent to the international data in terms of coverage (the first two curves being nearly identical) but including the Chinese data with the other international data significantly increases the coverage density. If this is the intent of the figure, this conclusion should be stated explicitly.
Author Response
Dear Professor,
Thank you very much for your constructive and helpful suggestions on our manuscript. As you pointed out in your comments, the manuscript needs to be revised and improved.
According to your suggestion, we have made detailed analysis and modification one by one. Please refer to the attachment for the detailed revision of the manuscript.
Thank you for your careful work.
Sincerely yours,
Yongjun Jia

Round 3
Reviewer 2 Report
The authors have incorporated the simple suggestions I made regarding language and clarity, but have ignored the more substantive comments. For example, in the discussion of the ASI algorithm on lines 321-329, I commented that obtaining different results does not necessarily imply that the new results are better. Their response was to add the word "obviously" to a conclusion that is not by any means obvious. Where is the evidence that the new results are more accurate? If they are not prepared to defend this conclusion, the authors should simply state the difference without claiming an improvement. Also, how does the new algorithm reduce the impacts of weather? (Not to mention that the impacts are presumably on the measurement, not on the phenomenon being measured.) By the way, I think that ASI stands for "Arctic Sea Ice" generally, but there are specific ASI algorithms, such as the one developed during the ARTIST project, and the specific algorithm should be referenced when making comparisons. For example, in the first use of the term, ASI should be defined parenthetically as simply (Arctic Sea Ice) but in subsequent references, when it is stated that the results of the new algorithm are "significantly lower than the ASI algorithm" the specific version should be specified, such as version 5.6i for example. Similarly, "the traditional ASI algorithm" should be replaced by a reference to the specific version.
Author Response
Thank you for your hard work! Please see the attachment.

Round 4
Reviewer 2 Report
No further comments.